#### CFD simulations to optimize the blades design of water wheels 1

- Emanuele Quaranta<sup>1</sup>, Roberto Revelli<sup>2</sup> 2
- 1,2 Department of Environment, Land and Infrastructure Engineering, Politecnico di Torino, Turin, Italy 3
- Correspondence to: Emanuele Quaranta (emanuele.quaranta@polito.it/quarantaemanuele@yahoo.it), Roberto Revelli
- (roberto.revelli@polito.it)
- Abstract. In low head sites and at low discharges, water wheels can be considered among the most convenient hydropower
- converters to install. The scope of this work is to improve the performance of an existing breastshot water wheel changing
- the blades shape, using Computational Fluid Dynamic (CFD) simulations. Three optimal profiles are investigated: the profile
- of the existing blades, a circular profile and an elliptical profile. The results are validated performing experimental tests on
- the wheel with the existing profile. The numerical results show that the efficiency of breastshot wheels is affected by the
- blades profile. The average increase in efficiency using the new circular profile is about 4% with respect to the profile of the
- existing blades.

#### 13 **1** Introduction

Electricity production in large scale from renewable energy sources has become an important purpose in the European 15 Commission legislations. Among renewable energy sources, hydropower is considered to be one of the most important ones 16 (Bódis et al., 2014). However, large hydropower plants need the construction of large dams, buildings and installations for 17 the generation, regulation and transmission of power, and the payback periods are generally long. In addition, there are often 18 many adverse effects and drawbacks on the ecosystems, for example the flooding of large areas and the interruption of the 19 continuity of the river. Micro-hydropower (net input power lower than 100 kW) is instead considered more eco-friendly. 20 Therefore, the interest in micro hydropower is increasing. Most of low head and low discharge sites are still not exploited, 21 since standard turbines cannot be employed economically in such conditions (Bozhinova et al., 2013; Müller and Kauppert, 22 2004).

23 In Bozhinova et al. (2013) a review of hydropower converters for very low heads has been presented, and an attractive 24 opportunity in micro hydro field can be represented by gravity water wheels. Gravity water wheels exploit the potential 25 energy of water and a portion of the kinetic energy. They can be classified in overshot wheels, where the water enters into 26 the wheel from the top, and breastshot water wheels, where the water fills the buckets entering from the upstream side of the 27 wheel. Breastshot water wheels can be divided in high, middle and low, depending whether the water entry point to the 28 wheel is over the rotation axis (in the uppermost third of the wheel), near the axis (in the middle third of the wheel) or under 29 the axis (in the lowest third of the wheel), respectively. In breastshot water wheels the upstream water level can be controlled 30 by inflow structures. When there is an overflow weir or an undershot weir (with a sluice gate to regulate the upstream water 31 level), breastshot wheels are called slow or fast, respectively, considering the higher flow velocity to the wheel occurring in the latter case (Quaranta and Revelli, 2016a). Low breastshot wheels for very low heads are called undershot wheels. 32 33 Zuppinger and Sagebien undeshot water wheels are used in sites with very low heads (typically less than 1.5 m), and the 34 upstream conditions can be controlled by an inflow weir, so that the approaching flow velocity is very low, generally less 35 than 1 m/s (Quaranta and Müller, 2017). A particular kind of fast undershot water wheel which exploits well the kinetic 36 energy of the water is the Poncelet wheel (Poncelet, 1843). Poncelet wheels are generally installed in straight channels, with 37 no bed drops or geometric heads through the wheel. The channel drop is present downstream of the wheel, so that the blades 38 do not interfere with the tailrace. The inflow is realized with a sluice gate which is very close to the wheel, in order to 39 increase the flow velocity. The water jet exchanges its momentum with the wheel flowing along the blades.

The maximum efficiency of water wheels can be higher than of 80% for overshot water wheels (Quaranta and Revelli, 41 2015b), 75% for breastshot water wheels (Quaranta and Revelli, 2015a, Quaranta and Revelli, 2016a, Vidali et al., 2016), 42 higher than 80% for undershot Zuppinger and Sagebien water wheels (Quaranta and Müller, 2017) and approximately 55% 43 for Poncelet wheels. Although water wheels are environmental friendly and efficient hydropower converters, only a small 44 amount of research has been spent on their performance characteristics in the last century. There are now some companies 45 and research centers which are currently dealing with water wheels, especially for electricity generation. Due to their several 46 advantages over turbines (lower costs, shorter payback period, higher and simpler adaptability to the external conditions, but 47 no simpler design), water wheels may constitute a suitable technology for the economic development, in particular in rural 48 areas and developing worlds.

# 49 1.1 Scope of the work

In Quaranta and Revelli (2015a) a theoretical model has been proposed to estimate the power losses of breastshot wheels, and in Vidali et al. (2016) a dimensional approach was performed. In Quaranta and Revelli (2016b) the number of the blades has been investigated for breastshot water wheels. Concerning instead the blades profile, the general criteria that should be taken into account in the blades design are well established (Quaranta and Revelli, 2015a), whereas numerical or experimental investigations on the optimal profile of fast breastshot wheels' blades can be rarely found. The general design criteria for the blades profile are:

(1) the relative entry stream velocity in the impact point should be directed as the blade inclination, in order to reduce theinflow power losses;

(2) the uplift of water downstream of the wheel and the outflow power losses should be minimized. Hence the blades
should exit at a normal angle with respect the free surface at the tailrace, or with a backward inclination in order to reduce
the drag;

(3) the blades length should be long enough or curved in order to avoid losses of water at the root of the blades.

Therefore, the scope of this paper is to investigate by Computational Fluid Dynamic (CFD) simulations the effect of the 63 blades profile on the performance of fast breastshot wheels. This is justified by the fact that, although the general criteria for 64 the blades profile are well established, it is not so clear if the blades profile generates significant effects on the performance 65 of this kind of wheel (as previously illustrated). Similar uncertainty has also been found for Poncelet wheels: in Weisbach 66 (1849) and Faibairn (1864) the circular shape is suggested, while in Bresse (1869) the Author says that the blades curvature 67 is a matter of indifference. This work is also led by the need to improve the performance of an existing wheel, acting on the 68 blades shape. The existing wheel with its original blades profile was simulated and then two different profiles were also investigated. The 1:2 scale physical model of this wheel with the original blades profile has also been installed in the 69 70 Hydraulics laboratory of Politecnico di Torino, both for studying in detail the performance of breastshot wheels (Quaranta 71 and Revelli, 2015a), and to validate the numerical model.

CFD simulations for gravity water wheels have been already successfully used in Quaranta and Revelli (2016b), where
 the performance of the present breastshot water wheel has been investigated through CFD simulations for different blades
 numbers, and for overshot water wheels (Quaranta, 2017).

### 75 2 Method

The investigated breastshot water wheel is a 1:2 scale model (Froude similarity) of an existing one, sited in Verolengo, 77 near Turin (Italy) (Fig.1); it is made of 32 blades and the diameter is 4 m. The scaled wheel is 2.12 m in diameter and the 78 width of the installed wheel is b=0.65 m. The channel which conveys water to the wheel is 0.67 m wide; 0.7 m upstream of

- the entry point to the wheel there is a sluice gate. The geometric head (or channel drop, which is the difference between the
- elevation of the channel's bed upstream and downstream of the wheel) is 0.35 m, thus the wheel is a low breastshot wheel.

84

Figure 1: The existing breastshot water wheels, whose 1:2 scale model was investigated in this work. The original diameter is 4 m.

86

Furthermore, since the water flow accelerates passing under the undershot weir (through the sluice gate opening) and the blades are shaped in order that, at the beginning of the filling process, the jet flows along them (before becoming at rest in the buckets), the inflow process is similar to *Poncelet* wheels. In Fig.2 the sketch of the scaled wheel is reported.

Figure 2: The sketch of the investigated breastshot water wheel, which is the 1:2 scale model of the existing one.

93

2.1 Blades profile

Three different shapes are here investigated by CFD analyses: the profile of the existing blades (1), a circular modified profile (2) and an elliptical profile (3) (Fig.3). The modified profiles (2) and (3) are designed with the same tip inclination of profile (1), which is 16° on the horizontal in the entry point, in order to compare objectively the effect of different profiles. The tip inclination of the profiles is almost parallel to the relative flow velocity, in order to minimize the impact power losses. In this case, the profiles also minimize the downstream power losses, since they exit from the tailrace approximately normally, without uplifting water. The angle between the profiles and the tailrace water surface is 83°; it is good to be smaller than 90°, since the slight backward inclination at the tip allows to reduce the drag.

102