# Peer review of "CFD simulations to optimize the blades design of water wheels 1"

_Drinking Water Engineering and Science, 2017_

## Referee Comment (RC1) · Anonymous Referee #1 · 15 Mar 2017

My major comment to this paper is that the experiments are not explained in detail, so that the interpretation of the values of the numerical simulations is not traceable. It is important to give details about how and with which measurement devices torque, rpm, flow and hydraulic heads are measured and to now the uncertainties of these devices. Are the measured values validated? It seems to me that the difference between the experimental and the numerical results might be within the uncertainties of the experimental setup and thus the impact of the different forms of the blades on the torque and with this on the efficiency and the power may not be really distinguishable. Please validate the measurement data and show the results in the paper.

My other comments are: line 15 Why is hydropower considered as one of the most important renewable energies? line 17 How long is a long payback time? line 17 Incorporate the EU Water Framework Directive (2000); this is the official document

[Figure]

**DWESD**

on which the continuity of rivers and streams is specified. line 20 ". . . . are still not exploited . . .." This needs to be considered more differentiated. There have been thousands of small mills up to about 100 years ago and then got neglected as turbines (that could also transform higher flows into electricity) and generators were invented. So, many of the sites have been exploited but are nowadays neglected. line 29 ". . . . The upstream water level can be controlled. . ..". why "can"? Are there other ways to control the water level? line 41 Why are water wheels environmental friendly? How do you define this? Is this proven? If so, please quote. line 93ff Why have you chosen exactly this curvature for the modified blade profiles? Is there any resemblance to other blade profiles e.g. Zuppinger Wheel blades? line 129: ". . ..an optimal radius can be considered. . .." maybe was considered is more correct? Did you utilize $r = 0{,}25$m? Or could it be another value? table 1 I am sorry, but I cannot reproduce some of the calculated values (namely -1,16%, 5,4% and 5,7%). Please consider: would it be more feasible to compare in column 7 (Cexp-C1)/C1, so that all percentages are investigated from the same basis? line 198/199 The values in the text are not identical with the values in table 1.

---

## Author Comment (AC1) · 16 Mar 2017

Dear Anonymous Reviewer, thank you very much for your comments. In agreement with your comments, we will modify the manuscript, that, we hope, will become better and easier. We answer to your comments point to point.
* * *
1)

My major comment to this paper is that the experiments are not explained in detail, so that the interpretation of the values of the numerical simulations is not traceable. It isimportant to give details about how and with which measurement devices torque, rpm,flow and hydraulic heads are measured and to now the uncertainties of these

devices.

Thank you very much for your suggestions. As now reported in chapter 2.2, we have performed detailed experimental tests, and we have deeply described the experimental results in Quaranta and Revelli (2015a) and Quaranta and Revelli (2016). In order to make the paper shorter and simpler, we have preferred to remind to these two publications, where all the instruments uncertainties have been described. In this paper we have added the torque uncertainty, which is the most important one to consider for the aim of the paper.
* * *
2)

Are the measured values validated? It seems to me that the difference between the experimental and the numerical results might be within the uncertainties of the experimental setup and thus the impact of the different forms of the blades on the torque and with this on the efficiency and the power may not be really distinguishable. Please validate the measurement data and show the results in the paper.

Thank you. As discussed in Quaranta and Revelli (2016b), the results are validated on the experimental torque (also reported in Table 1) and water depths. We will add to the present manuscript the torque uncertainty, which is 6 Nm. The torque difference generated by using different profiles might be within the uncertainties of the experimental setup. Anyway, being the numerical model the same for each configuration, the uncertainty of each result is reasonably the same. Therefore, it is possible to say that profile 2 is the best among the investigated profiles. The results on the torque may quantitatively differ from the real one (due to the approximation necessarily unavoidable by CFD simulations). However, being the discrepancy reasonably the same for each configuration, it is possible to say that the percentage torque differences among the different profiles well respect the real/exact ones. Thus profile 2 is the optimal profile among the investigated ones, and the efficiency of the wheel can be improved.

—————————————————————————————————————

3)

My other comments are: line 15 Why is hydropower considered as one of the mostim-
portant renewable energies?

Thank you. This because it is the most used renewable source in the world. We will
add a reference, specifying this aspect better.

—————————————————————————————————————

4) line 17 How long is a long payback time?

Thank you. Yes, we agree. We have written now that payback times of micro hydro are
shorter than payback times of larger hydro schemes.

—————————————————————————————————————

5)

line 17 Incorporate the EU Water Framework Directive (2000); this is the official docu-
ment on which the continuity of rivers and streams is specified.

Yes, thank you.

—————————————————————————————————————

6)

line 20 ": : :. are still notexploited : : :." This needs to be considered more differen-
tiated. There have beenthousands of small mills up to about 100 years ago and then
got neglected as turbines(that could also transform higher flows into electricity) and
generators were invented.So, many of the sites have been exploited but are nowadays
neglected.

Thank you for your specification. We will specify better.
* * *
7)

line 29 ": : :. The upstream water level can be controlled: : :". why "can"? Are there other ways tocontrol the water level?

In water wheels the upstream water level can be controlled/imposed by means of sluice gates or inflow weirs. However, these inflow structures are not mandatory.
* * *
8)

line 41 Why are water wheels environmental friendly? How doyou define this? Is this proven? If so, please quote.

They are environmental friendly because of the large cells and low rotational speed. We will quote what we say. Thank you.
* * *
9)

line 93. Why have you chosen exactly this curvature for the modified blade profiles? Is there any resemblance to other blade profiles e.g. Zuppinger Wheel blades?

Zuppinger water wheels are different from the investigated wheel. Zuppinger wheels do not have sluice gate upstream. The water level just upstream of the Zuppinger wheel is deeper. Zuppinger wheels do not exploit the kinetic energy; actually, they generate very high power losses when entering into the water (Quaranta and Muller, 2017). The investigated water wheel, instead, exploits the kinetic energy of the entering flow, although some power losses are obviously generated. This curvature was chosen because it well satisfies all the prescription described in section 1.1. As we say, the lower the curvature radius of the blades, the better the power output, since the

higher the changing in momentum of the water flow, i.e. the generated force. However, there is a limit on the minimum radius. For example, in our case, and considering the configuration in the entry point, a curvature radius of 0.2 m (1/5 R, where R is the wheel radius) would have a portion of the profile that would be vertical. This would generate separation of flow and resistance; the flow would tend to fall down during the filling process, with additional power losses. Therefore, the chosen circular radius is the minimum optimal one.
* * *
10)

line 129: ": : :.an optimal radius can be considered: : :." maybe was considered is more correct? Did you utilize r =0,25m? Or could it be another value?

Right. Thank you!
* * *
11) table 1 I am sorry, but I cannot reproduce some of the calculated values (namely -1,16%, 5,4% and 5,7%).

Yes, we agree, 1.11 instead of 1.16, and 5.5 instead of 5.7. The other values seem to be right to us (C1-Cexp)/Cexp.
* * *
12)

Please consider: would itbe more feasible to compare in column 7 (Cexp-C1)/C1, so that all percentages areinvestigated from the same basis?

Thank you, good osservation. We did (C1-Cexp)/Cexp, because this is the validation, where we compare the numerical torque with the experimental one. Then, we calculate the numerical differences of the two additional profiles with respect to the first one. We

will specify this in the text. Thank you
* * *
13)

line 198/199 The values in the text are not identicalwith the values in table 1.

Thank you.

Please also note the supplement to this comment:
http://www.drink-water-eng-sci-discuss.net/dwes-2017-2/dwes-2017-2-AC1-
supplement.pdf

---

## Short Comment (SC1) · 29 Mar 2017

My major concern is the water wheel technology. Water wheel has been used as a energy generation device for hundreds of years, why the authors still stick with this technology instead of using turbines and other advance hydro-kinetic devices? The authors have to convince the readers that (1) the presented water wheel is a competitive energy technology comparing to the advance hydraulic energy conversion technologies; and (2) the paper does bring some original contribution on the existing knowledge base. The authors mentioned that the water wheels have "several advantages over turbines" but did not support this statement with enough proofs.

I also have questions about the benchmark of the optimal design. The authors displayed three wheel profiles to compare, how the three profiles were selected? why

not other shapes (3D spiral etc.)? How the dimensions were determined? what if we change the radius of the circular profile or the curvature of the elliptical profile? It is not convincing that the listed profiles and dimensions would necessarily lead to the optimal performance.

From the results, by using the circular profile, the momentum increasing was only 5.6% but it may cause extra cost in fabricating water wheels with complex shapes, is it feasible to design and fabricate circular wheels for only 5.6% increment?

A complete research paper should include at least two ways to "proofreading" the obtained results. A section of experimental study and a detailed demonstration of experimental results and comparison of the experimental results to the numerical results are necessary!

The CFD simulation part needs more information, what software was used for modeling and simulation? what types of element (2D? 3D?) were used to mesh the model? What material properties were used for modeling the water/fluid and the water wheel/solid?

The topic of hydropower generation technology and water/paddle wheel design optimization has been visited by many researchers. The authors at least need to pay attention to some recent studies:

1. Akinyemi and Liu, "CFD modeling and simulation of a hydropower system in generating clean electricity from water flow", International Journal of Energy and Environmental Engineering, 6, 2015.

2. Akinyemi, Chambers, and Liu, "Evaluation of the power generation capacity of hydrokinetic generator device using computational analysis and hydrodynamic similitude", journal of power and energy engineering, 3, 2015.

3. Liu and Peymani, "Development and computational verification of an analytical model to evaluate performance of paddle wheel in generating electricity from moving fluid", Distributed generation and alternative energy journal, 30, 2015.

4. Liu and Peymani, "Evaluation of paddle wheels in generating hydroelectric power", IMECE2012-85121, IMECE 2012, Houston, TX, USA, 2012.

Some minor typos or grammar errors:

1. P4, ln 120, "manufacture process" should be "manufacturing process". 2. Ln 126, was the optimal radius 0.25 m randomly picked? 3. How were the dimensions of profile 2 and 3 determined? 4. Section 2.2 is duplicate, needs to be renumbered. 5. Pg 5, ln 162, should be "as shown in Fig. 4". 6. Ln 163, should be "different blade numbers". 7. Ln 202 "is to prefer to" should be "is preferred to".

---

## Author Comment (AC2) · 2 Apr 2017

Dear Reviewer prof. Liu, thank you very much for your suggestions and comments. We will answer to them in the following lines.

1)

My major concern is the water wheel technology. Water wheel has been used as a energy generation device for hundreds of years, why the authors still stick with this technology instead of using turbines and other advance hydro-kinetic devices? The authors have to convince the readers that (1) the presented water wheel is a competitive energy technology comparing to the advance hydraulic energy conversion technologies; and (2) the paper does bring some original contribution on the existing knowledge base. The authors mentioned that the water wheels have "several advantages over turbines"

but did not support this statement with enough proofs.

ANSWER

Thank you. We clarify that we are talking about gravity water wheels (undershot, breastshot and overshot), that are gravity machines and are different from hydro-kinetic devices. Hydro kinetic devices are used in flowing water and exploit the flow kinetic energy (like Darrius turbines or stream water wheels). Instead, gravity wheels are used in sites with small heads, and use the water weight to generate mechanical energy. The investigated wheel, instead, partially exploits also the kinetic energy, although the water weight is still the predominant driving force. Moreover, water wheels are safer for fish, since they do not have pressurized structures like conduits. They also need few engineering works, making them suitable for rural areas.

We have added the following information to answer to comment 1):

Due to their several advantages over turbines water wheels may constitute a suitable technology for the economic development, in particular in rural areas and lower-income countries . Indeed, the efficiency can be maintained constant over a wide range of external conditions, without acting on the pitch of the blades. The total cost of a water wheel depends on its dimensions and geometry. In Germany (Müller and Kauppert, 2002), overshot water wheels are currently built (including installation and grid connection) for 3900Åů4400 €kW. Undershot wheels cost 6900Åů8700 €kW, Archimedes screws approximately 7400Åů7800 €kW of installed capacity. For comparison, low head Kaplan turbines cost 13000Åů13900 €kW. Water wheels cost is between 33% and 66% of reaction turbines. Payback periods can be estimated as 14Åů16 for Archimedes screws, 7Åů9 years for an overshot and 12Åů17 years for an undershot wheel, (with expected life time of 30 years), which are very low if compared to Kaplan turbine installations, where 25 Åů30 years payback periods can be expected (Müller and Kauppert 2004).

Concerning with comment 2), in section 1.1 we have discussed the scope of the work.

It is not clear if the blade profile affects the performance of fast breastshot wheels. Therefore, our aim is to investigate this point, showing how the blade profile can affect the performance.

2)

I also have questions about the benchmark of the optimal design. The authors displayed three wheel profiles to compare, how the three profiles were selected? why not other shapes (3D spiral etc.)? How the dimensions were determined? what if we change the radius of the circular profile or the curvature of the elliptical profile? It is not convincing that the listed profiles and dimensions would necessarily lead to the optimal performance. From the results, by using the circular profile, the momentum increasing was only 5.6% but it may cause extra cost in fabricating water wheels with complex shapes, is it feasible to design and fabricate circular wheels for only 5.6% increment?

ANSWER

We have chosen the two shapes because they are well defined geometric shapes; this means that their fabrication process can be automated easily, also when water wheels are constructed by artisans, for example in rural areas and lower-income countries. Gravity water wheels are not axial machines and they are not immersed in water. Those curvatures were chosen to satisfy the three points explained in section 1.1. In particular: profile 2) (the circular) was chosen because the lower the curvature radius of the blades, the better the power output, since the higher the changing in momentum of the water flow, i.e. the generated force. However, there is a limit on the minimum radius. For example, in our case, and considering the configuration in the entry point, a curvature radius of 0.2 m (1/5 R, where R is the wheel radius) would have a portion of the profile that would be vertical. This would generate separation of flow and would increase the resistance; the flow would tend to fall down during the filling process, with additional power losses. Therefore, the chosen circular radius is an optimal one. Then, we have added an elliptical profile that satisfies the three points in section 1.1 in order

to compare the effect of changing the profile itself.

3)

A complete research paper should include at least two ways to "proofreading" the obtained results. A section of experimental study and a detailed demonstration of experimental results and comparison of the experimental results to the numerical results are necessary! The CFD simulation part needs more information, what software was used for modeling and simulation? what types of element (2D? 3D?) were used to mesh the model? What material properties were used for modeling the water/fluid and the water wheel/solid?

ANSWER

Thank you. The experimental results are well described in Quaranta and Revelli (2015 and 2016), so we have preferred to remind to these two papers in order to make this paper more concise and simpler. Concerning with the mesh, in section 2.2 we have described it, and now we have added the missing information. In section 2.3, we have written the water properties.

4)

The topic of hydropower generation technology and water/paddle wheel design optimization has been visited by many researchers. The authors at least need to pay attention to some recent studies.

ANSWER

Yes, thank you. We have added these references, considering, however, that we are dealing with gravity wheels not immersed in water, while paddle wheels are in flowing water and are partially immersed in it. Anyway, we think that the suggested additional papers can be useful to complete our paper and to show more information about CFD applied to water wheels in general. Thank you. We have also corrected the minor typos.

Please also note the supplement to this comment:
http://www.drink-water-eng-sci-discuss.net/dwes-2017-2/dwes-2017-2-AC2-supplement.pdf